# Towards Fair RAG: On the Impact of Fair Ranking in Retrieval-Augmented Generation

## Abstract

Many language models now enhance their responses with retrieval capabilities, leading to the widespread adoption of retrieval-augmented generation (RAG) systems. However, despite retrieval being a core component of RAG, much of the research in this area overlooks the extensive body of work on fair ranking, neglecting the importance of considering all stakeholders involved. This paper presents the first systematic evaluation of RAG systems integrated with fair rankings. We focus specifically on measuring the fair exposure of each relevant item across the rankings utilized by RAG systems (i.e., item-side fairness), aiming to promote equitable growth for relevant item providers. To gain a deep understanding of the relationship between item-fairness, ranking quality, and generation quality in the context of RAG, we analyze nine different RAG systems that incorporate fair rankings across seven distinct datasets. Our findings indicate that RAG systems with fair rankings can maintain a high level of generation quality and, in many cases, even outperform traditional RAG systems, despite the general trend of a tradeoff between ensuring fairness and maintaining system-effectiveness. We believe our insights lay the groundwork for responsible and equitable RAG systems and open new avenues for future research. We publicly release our codebase and dataset.

## 1 Introduction

In recent years, the concept of fair ranking has emerged as a critical concern in modern information access systems (Ekstrand et al., 2022). However, despite its significance, fair ranking has yet to be thoroughly examined in the context of retrieval-augmented generation (RAG) (Lewis et al., 2020; Asai et al., 2024), a rapidly advancing trend in natural language processing (NLP) systems (Kim et al., 2024). To understand why this is important, consider the RAG system in Figure 1, where a user asks a question about running shoes. A classic retrieval system might return several documents containing information from various running shoe companies. If the RAG system only selects the top two documents, then information from the remaining two relevant companies will not be relayed to the predictive model and will likely be omitted from its answer. The fair ranking literature refers to this situation as unfair because some relevant companies (i.e., in documents at position 3 and 4) receive less or no exposure compared to equally relevant company in the top position (Ekstrand et al., 2022).

Understanding the effect of fair ranking in RAG is fundamental to ensuring responsible and equitable NLP systems. Since retrieval results in RAG often underlie response attribution (Gao et al., 2023), *unfair exposure of content* to the RAG system can result in incomplete evidence in responses (thus compromising recall of potentially relevant information for users) or downstream representational harms (thus creating or reinforcing biases across the set of relevant entities). In situations where content providers are compensated for contributions to inference, there can be financial implications for the unfairness (Balan et al., 2023; Lyu et al., 2023; Henderson et al., 2023). Indeed, the fair ranking literature indicates that these are precisely the harms that emerge when *people* are searchers (Ekstrand et al., 2022), much less RAG systems, where the searchers are *machines*. RAG complicates these challenges since it often truncates rankings to much shorter lengths to fit the generator's limited context size (Bahri et al., 2020; Hofstätter et al., 2023; Kim et al., 2024), making equal exposure of relevant items even harder.

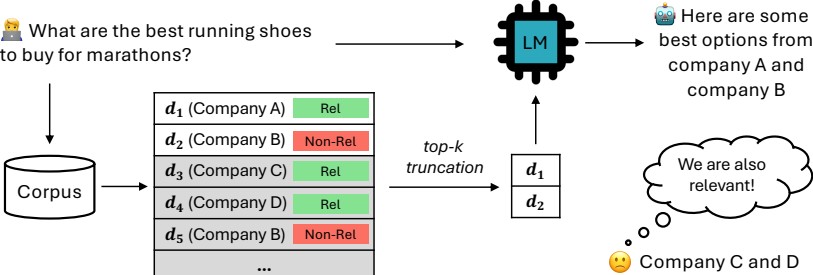

Figure 1: Fairness concerns in RAG. A simplified example of how RAG models can ignore equally relevant items ($d_3$ and $d_4$) and always consume the fixed top-scoring items ($d_1$ and $d_2$) with the same order of ranking over the multiple user requests. This is due to the deterministic nature of the retrieval process and a short context-length of a language model that necessitates the top-$k$ truncation of a ranked list.

Moreover, the fact that machines are now the searchers necessitates a different notion of item-worthiness (how deserving an item is to be included in a ranked list). Traditionally, ranking quality has been assessed based on relevance labels, which are created according to how relevant an item is to the user's query (Saracevic, 2016). However, with RAG systems, where the consumer is a language model, there is a growing shift towards evaluating ranking quality based on *utility labels*, which are determined by the usefulness of an item in aiding the model's task performance, rather than its relevance to the query (Salemi & Zamani, 2024a; Zhang et al., 2024a).

This shift from relevance to utility in the concept of item-worthiness can significantly alter our understanding of the relationship between fairness and ranking quality (Balagopalan et al., 2023)—particularly the tradeoffs that are well-known in the fair ranking literature (Biega et al., 2018; Diaz et al., 2020; Singh & Joachims, 2019). Since previous fair ranking studies were conducted based on relevance judgments, they may need to be reexamined in light of utility-based judgments within the context of RAG.

Our research aims to bridge the gap between traditional fair ranking studies and the emerging changes posed by RAG systems, ultimately enhancing our understanding of the interplay between fairness, ranking quality, and the effectiveness of RAG systems. We do this by evaluating RAG systems with a fairness-aware retriever across seven different tasks, experimenting with varying levels of retrieval fairness to observe changes in ranking quality and generation quality (utility).[1]

Our empirical results show that, in the context of machine users, there also exists an overall trend of fairness-quality tradeoff with respect to both retrieval and generation quality. However, the magnitude of this tradeoff is not particularly severe. In fact, we find that RAG models equipped with a fair ranker can often preserve a significant level of retrieval and generation quality, and in some cases, even surpass the quality achieved by the traditional RAG setup with a deterministic ranker that lacks fairness considerations.

This surprising finding offers significant insight into the potential of RAG-based applications, suggesting that fair treatment of individual content providers can be achieved without sacrificing much of the high-quality service delivered to end-users. This challenges the conventional assumption of an inevitable tradeoff between fairness and quality, opening new avenues for developing more equitable and effective RAG systems.

## 2 BACKGROUND & RELATED WORK

**Retrieval-Augmented Generation**. RAG, a specific type of retrieval-enhanced machine learning (REML) (Zamani et al., 2022; Kim et al., 2024), has been widely adopted in various domains, including language modeling (Khandelwal et al., 2020), question-answering (Izacard et al., 2023), personalization (Salemi et al., 2024b;a; Kumar et al., 2024; Neelakanteswara et al., 2024), and

---

[1]Throughout this paper, we use "utility" and "generation quality" interchangeably to refer to the downstream effectiveness of RAG models, measured by arbitrary string utility metrics.

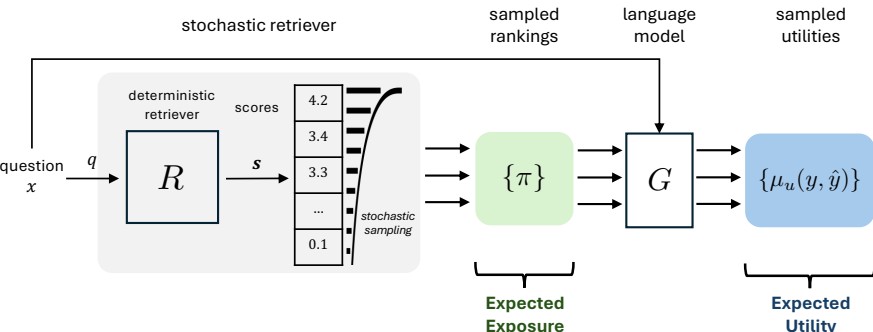

Figure 2: Experimental design to investigate the impact of item-fairness on ranking and generation quality in RAG. To evaluate system performance across multiple identical user requests, we sample $N$ rankings from a stochastic retriever. We then measure the fairness and quality of the sampled rankings (Expected Exposure) and assess the system's expected end-performance (Expected Utility). The query and prompt generators are omitted in the figure for brevity. Details on implementing a RAG system with fair rankings in a production environment can be found in Appendix E.

recommendation (Zeng et al., 2024). Studies on the evaluation of RAG models have primarily focused on their effectiveness, including end-to-end performance (Izacard et al., 2023; Guu et al., 2020; Lewis et al., 2020) and the assessment of individual components (Es et al., 2024; Saad-Falcon et al., 2024; Salemi & Zamani, 2024a). However, little research has focused on evaluating fairness in retrieval-enhanced generation models, with the exception of recent work (Shrestha et al., 2024), which improved demographic diversity in human image generation by conditioning a generative model with externally retrieved images that help debias the generation process.

**Fairness in Ranking**. Fair ranking has been approached through various definitions based on normative concerns, primarily with distinctions made according to the stakeholders we prioritize. These include consumer-side fairness (Mehrotra et al., 2017; Ekstrand et al., 2024), which focuses on how fairly a system delivers satisfaction to users; provider-side fairness (Sapiezynski et al., 2019; Jaenich et al., 2024), which addresses how fairly item providers receive monetary or reputational rewards; and item-side fairness (Jiang et al., 2024), which examines how fairly items are treated in terms of representation or exposure. The motivation of item-side fairness is closely linked to provider-side fairness, as unfair treatment of items can lead to unfair compensation for providers. These fairness concerns can be further categorized by the scope of stakeholders, encompassing individual fairness—ensuring similar treatment for similar individuals—and group fairness—ensuring equitable outcomes across different groups (Caton & Haas, 2020; Ekstrand et al., 2022). Previous studies have focused on developing metrics to measure fairness (Raj & Ekstrand, 2020) and optimizing fair retrievers within a single (Yang & Stoyanovich, 2017; Zehlike et al., 2017; Sapiezynski et al., 2019) or multiple rankings (Diaz et al., 2020; Singh & Joachims, 2018; Biega et al., 2018; Singh & Joachims, 2019). In the context of provider- and item-side fairness, ensuring equal exposure of similar items across multiple rankings has gained significant attention (Ekstrand et al., 2022). To achieve this, researchers have used stochastic rankers that return a distribution of rankings, in contrast to deterministic rankers commonly found in areas like RAG, which produce a fixed ranking. This approach ensures that, in expectation, similar items receive equal exposure across multiple user requests, with the distributions typically based on the merit of the rankings, such as an item's relevance (Diaz et al., 2020; Singh & Joachims, 2019).

In this research, we employ a stochastic ranker in RAG to enhance *individual item-side fairness*, aiming to ensure equal expected exposure for items that offer similar merits.

## 3 EXPERIMENTAL METHODOLOGY

In traditional RAG systems, a user input is used to query a retrieval system for recommended items from some corpus, which are then used for generation. Given user input $x$, a query $q$ generated by the query generation function $\phi_q(x)$, and a corpus of documents C, a *deterministic retriever* $\mathcal{R}(q, C)$ returns a fixed ranked list $L$ every time $q$ is seen. Retrieval is followed by a top $k$ truncation

which is passed to a prompt generation function $\phi_p(x, L_{1:k})$ that returns a final prompt $\bar{x}$, which is subsequently passed to the language model $\mathcal{G}(\bar{x})$. Because deterministic retrievers allocate exposure to the same item over repeated samples, RAG systems with deterministic retrievers present a challenge to ensuring equal exposure of relevant items to the generator.

To address the issue of unfairness in the rankings passed to the generator, we can convert a deterministic retriever into a stochastic retriever, which can, in expectation, provide fair rankings (Diaz et al., 2020). By sampling a ranking based on its quality to users—in this case generators—the expected exposure of different relevant items becomes similar and, therefore, fairer (Appendix E). Because decisions are stochastic, the fairness and quality of stochastic retrieval is evaluated based on a sample of rankings. Similarly, since a sampled ranking is processed by a generator, we also compute the expected generator effectiveness over sampled rankings. The complete evaluation pipeline of a RAG system with a stochastic retriever is illustrated in Figure 2.

The following sections describe how we construct a test collection with utility labels (§3.1), how we stochastically sample multiple rankings (§3.2), and how we evaluate the fairness and ranking quality of the sampled rankings (§3.3.1) and measure the effectiveness of a RAG system given multiple rankings (§3.3.2).

## 3.1 Construction of a Test Collection with Utility Labels

Setting an appropriate proxy for measuring item-worthiness is crucial in the evaluation of fairness (Balagopalan et al., 2023). Drawing on the insight that utility-based judgments are more suitable than relevance judgments in the context of RAG (Zhang et al., 2024a; Salemi & Zamani, 2024a), we annotate item-level utility labels for all items in the corpus.

We define an item's worthiness by the marginal gain in utility (utility-gain) it provides to a language model (specifically, the generator in a RAG system) when used to solve a specific task as part of the augmentation process. To assess this utility-gain, each item in the corpus is individually supplied to the generator along with an input question. The utility-gain is then calculated as the difference between the utility of the augmented generator and that of a baseline language model without any information about the item. Formally, let $u_i$ denote the baseline string utility score from the vanilla language model prompted only with the input question, and let $u_j$ represent the utility score from the language model with a prompt augmented by the $j$'th item $d_j$ in the corpus. The item $d_j$ is considered useful if the utility-gain $\delta_j = u_j - u_i$ is positive, and not useful otherwise (see Appendix B).

Therefore, the item-level utility labels are designed to be both task- and generator-dependent, as the utility of each item varies depending on the task and the language model used. This labeling process also aligns with the principles of task-based information retrieval, where, in the context of *human* searchers, document utility may vary on how the user expects to use the document (Kelly et al., 2013).

## 3.2 Fairness-Aware Stochastic Retriever

Stochastic retrievers have been used for various purposes, such as optimization of retrieval models (Bruch et al., 2020; Zamani & Bendersky, 2024; Guiver & Snelson, 2009; Oosterhuis, 2021), as well as ensuring equitable exposure of items (Oosterhuis, 2022; Diaz et al., 2020; Oosterhuis, 2021). Many of these studies use Plackett-Luce sampling (Plackett, 1975) to achieve the stochasticity of retrieval. We follow the line of research and formally define how we derive a fairness-aware stochastic retriever through Plackett-Luce sampling. To enhance sampling efficiency, we adopt the methodology of Oosterhuis (2021), and for controllable randomization, we utilize the approach proposed by Diaz et al. (2020).

Given $n$ items in a corpus C, a vector of retrieval scores $\mathbf{s} \in \mathbb{R}^n$ can be obtained from $\mathcal{R}(q, \mathbf{C})$, which can be used to generate a ranked list $L$. We then min-max normalize retrieval scores to be in $[0, 1]$ in order to construct a multinomial distribution over items (Biega et al., 2018). The probability of an item $d$ being selected as the $i$'th item in a new ranking $\pi$ through Plackett-Luce sampling is given by

$$p(d|L_{1:i-1}) = \frac{\exp(\bar{\mathbf{s}}_d)\mathbb{1}[d \notin L_{1:i-1}]}{\sum_{d' \in \mathbf{C} \setminus L_{1:i-1}} \exp(\bar{\mathbf{s}}_{d'})} \tag{1}$$

where $L_{1:i-1}$ is the partial ranking up to position $i - 1$, $\bar{\mathbf{s}}$ represents the normalized retrieval score vector, and $\bar{\mathbf{s}}_d$ is the normalized score of item $d$. Using this probability, we iteratively sample an

item, set its probability to 0, renormalize the distribution, and repeat the process. The probability of generating a complete ranking is then given by the product of the placement probabilities for each item, i.e., $p(\pi|q) = \prod_{i=1}^{n} p(\pi_i | \pi_{1:i-1})$.

This repeated sampling and renormalization process can be efficiently managed using the Gumbel-Softmax trick (Gumbel, 1954; Maddison et al., 2017), which enables the sampling of rankings to be performed at the speed of sorting (Oosterhuis, 2021). To do so, for each sampling iteration, we draw $U_i \sim \text{Uniform}(0, 1)$, followed by generating a Gumbel noise $G_i = -\log(-\log(U_i))$. The probability of each sampled ranking is then obtained by sorting the items based on their perturbed scores $\tilde{\mathbf{s}}_{d_i} = \bar{\mathbf{s}}_{d_i} + G_i$.

### 3.2.1 CONTROLLING THE LEVEL OF FAIRNESS

Adjusting the level of randomization directly controls the degree of item-fairness, aligning with our goal to observe how varying levels of fairness in rankings affect the ranking and generation quality of a RAG model. To obtain the controllability, we follow the work of Diaz et al. (2020) and use a temperature parameter $\alpha$. We apply the scalar $\alpha$ to each value in the normalized score vector $\bar{\mathbf{s}}$ by raising each value to the power of $\alpha$.[2] This process is done before the scores are passed to the sampling policy. Therefore, the modified sampling distribution is thus defined as:

$$p(d|L_{1:i-1}) = \frac{\exp(\bar{\mathbf{s}}_d^\alpha)\mathbb{1}[d \notin L_{1:i-1}]}{\sum_{d' \in C \setminus L_{1:i-1}} \exp(\bar{\mathbf{s}}_{d'}^\alpha)} \tag{2}$$

This implies that the sharpness of the sampling distribution is controlled by the $\alpha$. A higher $\alpha$ amplifies the probability of items with higher retrieval scores being sampled. Therefore, if multiple rankings are sampled by the stochastic retriever with high $\alpha$, it results in high disparity (i.e., item-side unfairness) of sampled rankings. At extreme, with considerably high $\alpha$, the procedure results in the identical rankings which is the behavior of a deterministic ranker (i.e., maximum item-unfairness). On the other hand, a lower $\alpha$ reduces the disparity of sampled rankings, making the exposure distribution fairer. At extreme, when $\alpha = 0$, the sampling procedure becomes uniformly random and achieves the lowest disparity (i.e., maximum item-fairness) in the sampled rankings.

## 3.3 EVALUATION

As mentioned in Section 3, because we are dealing with stochastic retrievers, we need to measure the *expected* behavior of the system. Let $\mathcal{S}(\mathbf{s}, N, k)$ be the stochastic sampler that samples a set of $N$ rankings $\sigma = \{\pi\}$, given the retrieval scores $\mathbf{s}$, where each ranking $\pi$ is truncated to the size of $k$. From each ranking, we can get an output $\hat{y} = \mathcal{G}(\phi_p(x, \pi))$. With an arbitrary fairness metric $\mu_f(\sigma)$ and a ranking quality metric $\mu_r(\sigma)$ that takes a set of rankings as an input, we can measure the degree of fairness and ranking quality of the sampled rankings. Similarly, an arbitrary string utility metric $\mu_u(y, \hat{y})$, such as ROUGE, can be used to assess an expected effectiveness of a RAG system by calculating the average of the $N$ metric scores.

In this paper, based on the empirical investigation done by Raj & Ekstrand (2020), we use expected exposure disparity (EE-D) and expected exposure relevance (EE-R) (Diaz et al., 2020) as $\mu_f$ and $\mu_r$, respectively (§3.3.1). For $\mu_u$, we select the metric depending on the task, and we get the expectation of the utility of a RAG model which we call an expected utility (EU) (§3.3.2).

### 3.3.1 EXPECTED EXPOSURE IN THE CONTEXT OF MACHINE USERS

Expected Exposure (EE) (Diaz et al., 2020) works by estimating the exposure of items across rankings (e.g., $\sigma$) created by a subject model, and comparing them with an optimal set of rankings that always satisfy the item-fairness. To represent the attention over $n$ items given by the consumer (generator in RAG), an $n \times 1$ system exposure vector $\epsilon$ is created. This is then compared with an $n \times 1$ target exposure vector $\epsilon^*$, where it represents the exposure of items allocated by an oracle retriever that always rank useful items above non-useful ones (Diaz et al., 2020).

---

[2]We normalized the values to the range of $[1, 2]$ instead of $[0, 1]$. The addition of 1 effectively serves the same purpose as adjusting a real-numbered $\alpha$. We chose this range to allow for an integer-valued $\alpha$.

With the system and target exposure vector $\epsilon \in \mathbb{R}^n$ and $\epsilon^* \in \mathbb{R}^n$, we can get the difference between the two by the squared $l2$ distance:

$$\|\epsilon - \epsilon^*\|_2^2 = \|\epsilon\|_2^2 - 2\langle \epsilon, \epsilon^* \rangle + \|\epsilon^*\|_2^2 \tag{3}$$

This difference yields two metrics useful for fairness and ranking quality evaluation. $\|\epsilon\|_2^2$ can be a measure for disparity of rankings (EE-D), and $\langle \epsilon, \epsilon^* \rangle$ can be a measure of ranking quality (EE-R) by calculating the degree of alignment of system exposure to the target exposure (i.e., how much of the exposure is on useful items). Therefore, the higher the value of EE-D, the more unfair the set of rankings are, and the higher the value of EE-R, the closer the set of system rankings are to the optimal set of rankings with respect to the ranking quality.

The exposure of an item is calculated by modeling users' (e.g., generators in RAG) attention to each item in a ranking. For example, one can assume that the user is affected by position bias and gives attention following an exponential decay (Moffat & Zobel, 2008). However, these browsing models were developed for human-users not for machine-users, so we need a different user behavior model for generators in RAG. For simplicity, we assume that the machine-user can consume the items by giving equal attention to all the items that were passed to the context, but pays 0 attention to the items placed after the $k$'th position due to the top-$k$ truncation. This makes the user browsing model a step function parameterized by $k$. In this work, a relevance-independent machine-user model (MU) is set to the step function that reflects the behavior of *top-k* truncation of RAG:

$$\text{MU}(i) = \begin{cases} 1 & \text{if } i \leq k \\ 0 & \text{otherwise} \end{cases} = \mathbb{1}[i \leq k] \tag{4}$$

Given this machine user browsing model and a mapping from item index to its rank denoted as $\bar{\pi}_d$, a system exposure for each item $d$ is calculated as

$$\epsilon_d = \sum_{\pi \in S_n} p(\pi|q)\text{MU}(\bar{\pi}_d) \tag{5}$$

and target exposures for a useful item $d$ and a unuseful item $d^-$ are calculated as

$$\epsilon_d^* = \frac{1}{m}\sum_{i=1}^{m}\text{MU}(i) = \begin{cases} 1 & \text{if } m \leq k \\ \frac{k}{m} & \text{otherwise} \end{cases} \qquad \epsilon_{d^-}^* = \begin{cases} \frac{k-m}{n-m} & \text{if } m \leq k \\ 0 & \text{otherwise} \end{cases} \tag{6}$$

### 3.3.2 EXPECTED UTILITY

Given the set of $N$ sampled rankings $\sigma$, we individually augment the generator with each ranking $\pi \in \sigma$, resulting in $N$ outputs from the generator. The utility of these outputs is then measured using an arbitrary string utility metric $\mu_u$. To determine the anticipated utility of a RAG model with fair rankings—represented by the tuple of a stochastic ranking sampler $\mathcal{S}$ and a generator $\mathcal{G}$—we calculate the expected utility (EU) of the RAG system given an instance $x$.

$$\text{EU}(\langle \mathcal{S}, \mathcal{G} \rangle | x) = \mathbb{E}_{\pi \sim \mathcal{S}}[\mu_u(y, \hat{y}_\pi)] = \sum_{\pi \in S_n} p(\pi|q)\mu_u(y, \hat{y}_\pi) \approx \frac{1}{N}\sum_{\pi \in \sigma} \mu_u(y, \hat{y}_\pi) \tag{7}$$

where $\hat{y}_\pi$ is the prediction of a system given the ranking $\pi$, $S_n$ is the symmetric group of a ranked list $L$ from the deterministic retriever $\mathcal{R}$, and $\sum_{\pi \in S_n} p(\pi|q) = 1$.

### 3.3.3 NORMALIZATION OF METRICS

From Equation 3, we decompose the metric into EE-D and EE-R. Since the bounds of these metrics depend on the number of useful items, normalization must be applied per query. Both metrics are min-max normalized based on their theoretical lower and upper bounds. We denote the normalized EE-D and EE-R as $\overline{\text{EE-D}}_q$ and $\overline{\text{EE-R}}_q$, respectively.

However, theoretically determining the bounds of the expected utility (EU) of a RAG model is challenging. To address this, we normalized the EU by the model's empirical upper bound, the maximum observed utility across all runs of the experiment with the same generators. To approach the true upper bound, these runs include RAG models with an oracle retriever that consistently ranks useful items (i.e., those with positive utility labels) above non-useful ones, stochastically returning

one of the $m!(n-m)!$ different rankings, where $m$ represents the number of useful items in the corpus. We denote the normalized EU as $\overline{\text{EU}}_q$, which can be interpreted as the distance to the optimal utility. From this section onwards, we omit the symbol $q$ from the normalized metrics for brevity. Proofs and details on how each metric is normalized by its lower and upper bounds can be found in the Appendix. C.

## 4 EXPERIMENT SETUP

We choose the LaMP benchmark (Salemi et al., 2024b) for our dataset. It assesses the personalization capability of language models through retrieval-augmentation of users' interaction history in a platform. LaMP includes various prediction tasks, such as classification, regression, and generation, and is well-suited for tasks where multiple items can be relevant/useful, unlike QA tasks with typically one or two provenance items. The retrieval items in LaMP have clear providers and consumers, aligning with our goal to ensure fairness for individual item providers. For example, in LaMP-1, retrieval items are academic papers, where exposure can increase citation counts for authors. In LaMP-4, retrieval items are news articles, where exposure can lead to monetary compensation for journalists. Due to the absence of a test set, we constructed a test collection as described in §3.1, using the first thousand entries of a user-based development set. Then, we discarded entries that have only one useful item in the corpus, as it is unnecessary to concern item-fairness in that case. We release the test collection, and the dataset statistics can be found in the Appendix J.

We use BM25 (lexical retriever) (Robertson et al., 1995), SPLADE (learned sparse retriever), and Contriever (bi-encoder dense retriever) (Izacard et al., 2022) as deterministic retrievers providing retrieval scores to base the sampling on. These models represent commonly used retrievers in the RAG literature (Kim et al., 2024). We use a sampling size of $N = 100$ and a truncation size of $k = 5$.

For generation models, we use Flan-T5-Small, Flan-T5-Base, and Flan-T5-XXL (Chung et al., 2022). For decoding strategy, beam size is set to 4, and no sampling strategy is used. This is to ensure that stochasticity is only introduced to the retriever for controlled experiments. Full implementation details can be found in Appendix K. With the three base retrievers and three generators, we configure nine different RAG models and evaluate them on the seven LaMP tasks. Utility measurement of the generated strings followed the metrics used in the LaMP paper.

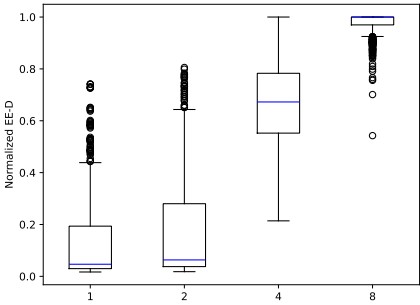

Figure 3: Effect of a temperature parameter $\alpha$ on the disparity of rankings, in the LaMP task 4, a text generation task. The RAG model is configured with the Contriever and Flan-T5-Base. Each data point represents the normalized EE-D of each run of the experiment (i.e., one query –>$N$ sampled rankings –> $\overline{\text{EE-D}}$ of the $N$ rankings).

We repeat the experiments with four different temperature parameters $\alpha = 1, 2, 4, 8$, which allows us to assess the utility of the RAG models with different levels of item-fairness. From Figure 3, we observe how effectively $\alpha$, described in the Equation 2, controls the disparity of rankings. For example, when $\alpha$ is set to 4, we usually obtain a set of sampled rankings with $\overline{\text{EE-D}}$ mostly in the range of [0.5, 0.8], and when $\alpha$ is set to 8, we often get a set of sampled rankings with $\overline{\text{EE-D}} = 1$. Refer to Appendix D to see the full description of the effect of $\alpha$ on the other metrics.

## 5 RESULTS

***RQ1****: Is there a tradeoff between ensuring item-fairness in rankings and maintaining high ranking quality when utility labels are used for evaluation?*

By gathering all four repeated runs of the experiments with different $\alpha$ values, we can plot the trend of ranking quality ($\overline{\text{EE-R}}$) against item fairness ($\overline{\text{EE-D}}$), as shown in Figure 4.

As shown in previous studies (Singh & Joachims, 2019; Diaz et al., 2020), there is a well-known tradeoff between fairness and ranking quality for human users. Similarly, we observe a general tradeoff for machine users. However, unlike past findings, this tradeoff is not always strict. For

instance, in Figure 4, both SPLADE and Contriever maintain consistently high ranking quality while being considerably fairer, and for BM25, ranking quality even improves as fairness increases, up to a certain point.

At the rightmost side of the lines, where $\overline{\text{EE-D}} = 1$ (representing the performance of deterministic rankers), we observe that these rankers do not always deliver the highest ranking quality. This suggests that commonly used deterministic rankers in RAG systems may be suboptimal, and that ranking quality can be improved while ensuring item fairness. This becomes even clearer when examining the impact of fair ranking on the downstream performance of a RAG system.

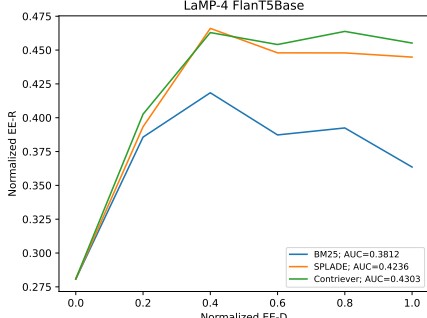

Figure 4: Relationship between item-fairness and the quality of rankings.

The leftmost side of the lines, where $\overline{\text{EE-D}} = 0$, represents the performance of a uniformly random ranking policy. At this point, the measured ranking quality should approximate the proportion of positively labeled items in the corpus, which is 31% based on data statistics (Appendix J). This is notably higher than in non-RAG (human-user) settings, where the percentage of relevant documents is typically much smaller, resulting in a $\overline{\text{EE-R}}$ value near 0 (Diaz et al., 2020).

To quantify the performance of fair rankers, we calculate the area under the disparity-ranking quality curve (Figure 4), with higher values indicating stronger ranking quality. We also measure the tradeoff by fitting a linear line to the experiment results, where a steeper slope reflects a stronger tradeoff between fairness and ranking quality. Based on these metrics, we observe that Contriever-based models exhibit the highest tradeoff, while BM25-based models show the lowest, despite their poor retrieval quality. Overall, SPLADE-based models achieve high retrieval quality while maintaining a relatively low tradeoff. For detailed plots and quantifications, refer to Appendix F.

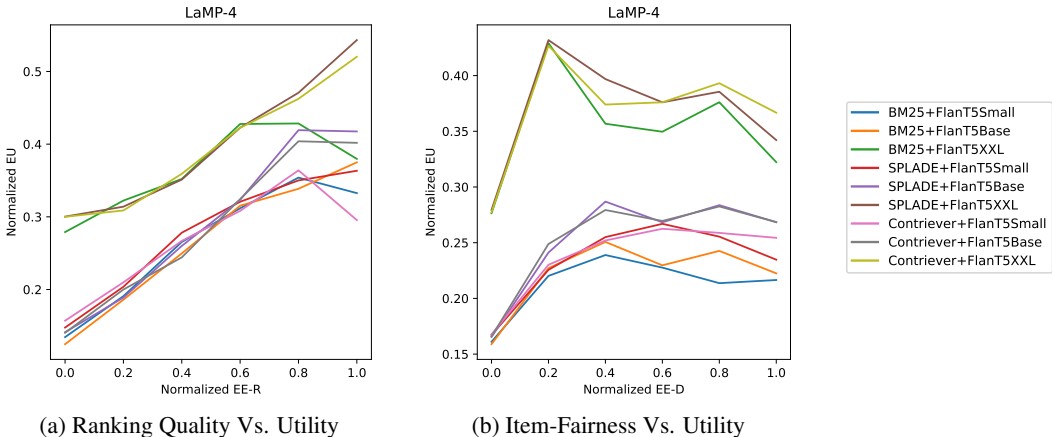

(a) Ranking Quality Vs. Utility        (b) Item-Fairness Vs. Utility

Figure 5: (a) Strong correlation between ranking quality and generation quality. (b) Fairness-generation quality tradeoff. Full plots and quantifications are provided in Appendix G and H.

| Model (baseline utility) | Fairness Intervals | | | | | Model (baseline utility) | Fairness Intervals | | | | |
|---|---|---|---|---|---|---|---|---|---|---|---|
| | [0.0, 0.2) | [0.2, 0.4) | [0.4, 0.6) | [0.6, 0.8) | [0.8, 1.0) | | [0.0, 0.2) | [0.2, 0.4) | [0.4, 0.6) | [0.6, 0.8) | [0.8, 1.0) |
| **LaMP-1** | | | | | | **LaMP-4** | | | | | |
| BM25+FlanT5Small (0.308) | -0.12 | -0.13 | -0.18 | -0.02 | -0.15 | BM25+FlanT5Small (0.217) | -0.06 | 0.00 | +0.02 | +0.01 | 0.00 |
| BM25+FlanT5Base (0.670) | -0.20 | -0.04 | -0.08 | -0.05 | -0.02 | BM25+FlanT5Base (0.223) | -0.06 | 0.00 | +0.03 | +0.01 | +0.02 |
| BM25+FlanT5XXL (0.531) | -0.07 | +0.03 | +0.02 | +0.06 | +0.11 | BM25+FlanT5XXL (0.322) | -0.05 | +0.11 | +0.03 | +0.03 | +0.05 |
| SPLADE+FlanT5Small (0.241) | -0.03 | -0.22 | +0.19 | -0.04 | +0.14 | SPLADE+FlanT5Small (0.235) | -0.07 | -0.01 | +0.02 | +0.03 | +0.02 |
| SPLADE+FlanT5Base (0.646) | -0.15 | +0.06 | +0.08 | 0.00 | +0.03 | SPLADE+FlanT5Base (0.268) | -0.10 | -0.03 | +0.02 | 0.00 | +0.02 |
| SPLADE+FlanT5XXL (0.671) | -0.18 | -0.16 | +0.05 | +0.02 | +0.01 | SPLADE+FlanT5XXL (0.342) | -0.06 | +0.09 | +0.05 | +0.03 | +0.04 |
| Contriever+FlanT5Small (0.286) | -0.08 | -0.29 | -0.06 | +0.03 | -0.14 | Contriever+FlanT5Small (0.254) | -0.09 | -0.02 | 0.00 | +0.01 | 0.00 |
| Contriever+FlanT5Base (0.637) | -0.16 | +0.05 | -0.06 | +0.03 | 0.00 | Contriever+FlanT5Base (0.268) | -0.10 | -0.02 | +0.01 | 0.00 | +0.01 |
| Contriever+FlanT5XXL (0.651) | -0.19 | -0.04 | -0.11 | +0.03 | 0.00 | Contriever+FlanT5XXL (0.367) | -0.09 | +0.06 | +0.01 | +0.01 | +0.03 |

Table 1: Each value in the table is the difference between the utility of a baseline (deterministic) RAG model and the average utility of a fairer RAG model at a specific fairness interval. Nonnegative differences are highlighted. Full results are listed in Appendix I.

***RQ2****: Is there a tradeoff between ensuring item-fairness in ranking and maintaining high generation quality of a RAG model?*

Before examining the relationship between fairness and RAG utility, Figure 5a shows an auxiliary result confirming a strong correlation between utility-based ranking quality and the effectiveness of RAG models. This is unsurprising, as item-worthiness judgments were based on the utility-gain provided by the generator. However, this correlation suggests that the tradeoff observed in the disparity-ranking quality curve (Figure 4) is likely to manifest similarly due to this strong relationship.

In fact, as observed from the disparity-utility curve (Figure 5b), we see a global trend of a non-strict tradeoff (i.e., RAG models maintain high generation quality while being considerably fair, and often even achieve higher quality).

However, a closer look at the local trend offers a significant insight: *RAG systems with fair ranking can often achieve higher system-effectiveness compared to models with deterministic rankers*. In Table 1, we divided the fairness levels into five intervals based on the normalized EE-D. As shown in the table and Appendix I, improving fairness to the level of $\overline{\text{EE-D}} \in [0.8, 1.0)$, and even $\overline{\text{EE-D}} \in [0.6, 0.8)$, can often enhance the utility of many RAG models across most LaMP tasks. For example, having $\overline{\text{EE-D}}$ in the range of $[0.8, 1.0)$ outperforms the baseline for all models in LaMP-2 and for seven out of nine models in LaMP tasks 4, 5, and 6.

## 6 DISCUSSION AND CONCLUSION

**Why do we often see higher system utility in fairer RAG models?** Although there is a general trend of a fairness-utility tradeoff, we observe that certain levels of fairness can actually improve the utility of a baseline RAG model. Recent line of research have uncovered relevant findings: 1) generators are not robust to changes in the position of useful information (Liu et al., 2024); 2) items with high retrieval scores often include distracting content that can reduce the system-effectiveness (Cuconasu et al., 2024; Ru et al., 2024); and 3) introducing some random documents can significantly boost the utility of RAG (Cuconasu et al., 2024).

Building on these existing results, we find that perturbing the initial ranking through stochastic sampling often can impact the performance of certain inference decisions and lead to changes in the system's expected end-performance. In our experiments, we observe that the expected utility generally increases within the fairness interval of [0.8, 1.0). This suggests that a fixed ranking from a deterministic ranker may be suboptimal for the generator, and that perturbing the ranking, along with the repositioning of items, not only improves expected end-performance but also enhances the fairness of the rankings.

Moreover, in fairness intervals where the system's expected utility improves, it is possible that either fewer distracting items were included in the ranking passed to the generator or useful, previously overlooked items (which may have been considered random) were introduced due to the ranking perturbation. However, while higher utility paired with increased item-fairness (even within fairness intervals as low as [0.4, 0.6)) may seem advantageous, practitioners should exercise caution. This could result in compensating providers of items irrelevant to user requests, particularly in scenarios where content providers are rewarded for contributing to inference outcomes.

**Machine-user browsing model**. Developing more sophisticated machine-user browsing models will result in more consistent and accurate evaluations of item-side fairness in RAG models, as the exposure of each item is influenced by the attention allocated by the machine-user. Initial research can draw inspiration from Liu et al. (2024), who found that LLMs tend to allocate more attention to the beginning and end of a context, with less focus on the middle. This line of inquiry aligns with the broader effort to create search engines tailored for machine-users (Salemi & Zamani, 2024b), specifically focusing on fairer search engines in this context. It should involve studying how LMs attend to each retrieved result within a context, analogous to how traditional search engine research models human browsing behavior (Dupret & Piwowarski, 2008).

**Measurement of string utility**. In line with the recent call for evaluating various valid output strings (Zhang et al., 2024b), we recognize the need for a similar approach to better measure system utility across different rankings given. Recall that our experiments were designed to provide the generator with different rankings for the same query, leading to varied outputs. This approach is motivated by

the idea that items not appearing in the top positions of deterministic rankings may still hold value and should be fairly considered by the system. In this context, the diverse outputs generated from different rankings may still be valid. However, we currently rely on a single target output string for comparison with predictions. Future work could focus on calculating the utility of diffuse predictions, enabling a more nuanced evaluation.

**Limitations**. We acknowledge that the evaluation cost of fair RAG systems can be high due to repeated sampling and inference steps. However, in production, only a single ranking is sampled, minimizing the impact on system latency (Appendix E). Also, a limitation in our utility labeling is that it considers single items, while multiple items may yield contrasting utility gains. Despite this, the strong correlation between ranking quality and system effectiveness suggests this approach reasonably approximates item-worthiness for evaluating the impact of fair ranking on RAG systems.

**Conclusion**. This study highlights the impact of fair rankings on both the ranking and generation quality of RAG systems. Through the extensive analysis, we show that fairer RAG systems not only maintain high generation quality but can also outperform traditional RAG models, challenging the notion of a strict tradeoff between fairness and effectiveness. Our findings provide valuable insights for developing responsible and equitable RAG systems and pave the way for future research in fair ranking and retrieval-augmented generation. In future work, we hope to extend this framework to consider graded or missing judgments and exploring the different notions of fairness in RAG systems, ultimately advancing the field of trustworthy RAG systems research.

## ETHICS STATEMENT

This study does not involve human subjects, harmful insights, or methodologies that raise ethical concerns. Additionally, there are no privacy, security, or legal issues associated with this work. Instead, this research follows existing work on fair ranking (Ekstrand et al., 2022) that aims to promote equity while acknowledging the potential limitations of technical opertaionalizations of fairness.

## REPRODUCIBILITY STATEMENT

The authors have made significant efforts to ensure the reproducibility of this research. A well-documented anonymized code repository containing all necessary materials to reproduce the experiments is available at `https://anonymous.4open.science/r/fair-rag-iclr-anonymous-9C09`. This repository describes a clear dataset source, a complete data generation pipeline using the LaMP benchmark (as detailed in §3.1 and Appendix B), and scripts for running the experiments. The code repository covers all stages of the research, including deterministic retrieval computation, stochastic retrieval, RAG model implementation, Expected Exposure and Expected Utility evaluation, and normalization of metric values following the theoretical proofs in Appendix C. For the part where the randomization is included a random seed is set to 42 across all experiments, and no sampling is used for language model decoding (as described in §4). This approach was implemented not only to ensure reproducibility but also to limit the introduction of stochasticity to the retrieval process, ensuring more reliable experiments. Implementations of nine RAG models were based on the publicly available retrievers and generators which are referenced in Appendix K.

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

# A NOTATION

| Notation | Description |
| --- | --- |
| $x$ | input instance |
| $y$ | output target |
| $\phi_q(x)$ | query generation function |
| $q$ | query returned by $\phi_q(x)$ |
| $d$ | retrieval item (document) |
| C | stored retrievable items (corpus) |
| $n$ | the number of $d$'s in C |
| $m$ | the number of useful items in C |
| | |
| $\mathcal{R}(q, \mathbf{C})$ | deterministic retriever |
| $L$ | ranked list returned by $\mathcal{R}(q, \mathbf{C})$ |
| $\mathbf{s} \in \mathbb{R}^n$ | retrieval scores returned by $\mathcal{R}(q, \mathbf{C})$ |
| $N$ | sampling size for stochastic sampling |
| $k$ | the number of $d$'s to retrieve per ranking |
| $\mathcal{S}(\mathbf{s}, N, k)$ | stochastic ranking sampler |
| $\sigma$ | set of $N$ sampled rankings returned by $\mathcal{S}(\mathbf{s}, N, k)$ |
| $\pi$ | sampled ranking $\in \sigma$ |
| $\phi_p(x, \pi)$ | prompt generation function |
| $\overline{x}$ | prompt returned by $\phi_p(x, \pi)$ |
| $\mathcal{G}(\overline{x})$ | language model |
| $\hat{y}$ | predicted output from $\mathcal{G}(\overline{x})$ |
| | |
| $wor(d\|x)$ | worthiness of an item $d$ given an input $x$ |
| $\mu_f(\sigma)$ | fairness metric of rankings |
| $\mu_r(\sigma)$ | relevance metric of rankings |
| $\mu_u(y, \hat{y})$ | string utility metric |
| $\epsilon \in \mathbb{R}^n$ | expected exposure of all items in C |
| $\epsilon^* \in \mathbb{R}^n$ | target exposure of all items in C |

Table 2: Notation.

## B LABELING PROCEDURE

**Algorithm 1** Labeling Procedure of Binary Utility Labels

1: $\mathcal{D} = \{(x_1, y_1), (x_2, y_2), \cdots, (x_T, y_T)\}$       $\triangleright$ dataset of size $T$
2: **for** $(x_i, y_i) \in \mathcal{D}$ **do**
3:     $u_i \leftarrow \mu_u(y_i, \mathcal{G}(x_i))$       $\triangleright$ string utility of a baseline model without augmentation
4:     **for** $d_j \in \mathbf{C}$ **do**
5:       $\hat{y}_j \leftarrow \mathcal{G}(\phi_p(x_i, d_j))$
6:       $u_j \leftarrow \mu_u(y_i, \hat{y}_j)$       $\triangleright$ string utility of a generator augmented with one item
7:       $\delta_j \leftarrow (u_j - u_i)$       $\triangleright$ marginal utility gained from the augmentation
8:       $wor(d_j|x_i) \leftarrow 0$
9:       **if** $\delta_j > 0$ **then**       $\triangleright$ binary decision of item-worthiness by the utility-gain
10:         $wor(d_j|x_i) \leftarrow 1$
11:       **end if**
12:     **end for**
13: **end for**

# C    NORMALIZATION OF METRICS

## C.1    NORMALIZATION OF EE-D

The disparity measure EE-D should be normalized by its true upper and lower bound.

**Theorem 1.** $\|\epsilon\|_2^2 \in [0, \|\bar{\epsilon}\|_2^2]$, where $\bar{\epsilon}$ is an exposure vector derived from any deterministic ranking.

*Proof.* The lower bound is achieved by a uniform random policy. Each item $d$ will have exposure of $\frac{1}{n}$. However, it is reasonable to assume that it is approximately 0, since the size of most of the retrieval corpus is very large. Also, it is common that the corpus consists of majority of non-relevant items. The implication is that, for the optimal exposure, since the $n - m$ non-relevant items are shuffled amongst themselves, each will have an expected exposure of close to 0. Thus, assuming large $n$ and relatively small $m$,

$$\frac{1}{n} < \frac{1}{n-m} \approx 0 \tag{8}$$

For upper bound, recall that the $\epsilon$ is computed based on samples of rankings from a stochastic policy. For relevance-independent browsing models (e.g., MU), all rankings $\pi \in S_n$ have identical exposure norms $\|\epsilon^\pi\|_2^2$, where $\epsilon^\pi$ is the exposure of items for ranking $\pi$ sampled from the stochastic policy. Then,

$$\|\epsilon\|_2^2 = \|\mathbb{E}_\pi[\epsilon^\pi]\|_2^2 \tag{9}$$

$$\leq \mathbb{E}_\pi[\|\epsilon^\pi\|_2^2] \tag{10}$$

$$= \mathbb{E}_\pi[\|\bar{\epsilon}\|_2^2] = \|\bar{\epsilon}\|_2^2 \tag{11}$$

**Corollary 1.** *For machine user browsing model with top-k consumption and equal attention to the top items, $\|\epsilon\|_2^2 \in [0, k]$*

With MU, the upper bound of EE-D, $\|\bar{\epsilon}\|_2^2$ becomes $\sum_{i=1}^n \text{MU}(i)^2 = k$. Therefore, per query $q$, we calculate a normalized EE-D

$$\overline{\text{EE-D}}_q = \|\epsilon\|_2^2/k \quad \in [0,1] \tag{12}$$

## C.2    NORMALIZATION OF EE-R

The ranking quality measure EE-R should be normalized by its true upper and lower bound.

**Theorem 2.** $\langle \epsilon, \epsilon^* \rangle \in [0, \|\epsilon^*\|_2^2]$

*Proof.* The lower bound is achieved when $\epsilon$ becomes $\epsilon^-$, which is an exposure vector of the worst case ranking $\pi^-$ (permutations that rank all non-relevant items above relevant items). Given the assumption made from equation 8 and, $C^+$ and $C^-$, which are set of indices of relevant and non-relevant items, respectively,

$$\langle \epsilon^-, \epsilon^* \rangle = \sum_{i=1}^n \epsilon_i^- \epsilon_i^* \tag{13}$$

$$\approx \sum_{i \in C^+} 0\epsilon_i^* + \sum_{i \in C^-} \epsilon_i^- 0 = 0 \quad \text{(from 8)} \tag{14}$$

Intuitively, the upper bound is achieved when $\epsilon$ becomes $\epsilon^*$, thus $\langle \epsilon^*, \epsilon^* \rangle = \|\epsilon^*\|_2^2$. Alternatively, we can show that any convex combination of optimal rankings will have a $\langle \epsilon, \epsilon^* \rangle = \|\epsilon^*\|_2^2$.

Let $\epsilon_d^*$ be the exposure of a relevant items, $S_n^*$ be the set of all optimal rankings, $w_{\pi'}$ be the weight on $\pi \in S_n^*$ such that $\sum_{\pi \in S_n^*} w_{\pi'} = 1$, and $\epsilon'$ be the exposure vector associated with $\pi'$.

$$\langle \epsilon, \epsilon^* \rangle = \sum_{i=1}^{n} \epsilon_i \epsilon_i^* = \sum_{i \in C^+} \epsilon_i \epsilon_i^* \qquad \text{(from 8)} \tag{15}$$

$$= \epsilon_d^* \sum_{i \in C^+} \epsilon_i \qquad \text{(equal exposure principle)} \tag{16}$$

$$= \epsilon_d^* \sum_{i \in C^+} \sum_{\pi' \in S_n^*} w_{\pi'} \epsilon_i' \tag{17}$$

$$= \epsilon_d^* \sum_{\pi' \in S_n^*} w_{\pi'} \sum_{i \in C^+} \epsilon_i' \tag{18}$$

$$\leq \epsilon_d^* \sum_{\pi' \in S_n^*} w_{\pi'}(m\epsilon_d^*) \qquad \text{(since } \epsilon' \text{ is optimal)} \tag{19}$$

$$= \epsilon_d^*(m\epsilon_d^*) \qquad \text{(since } \sum_{\pi \in S_n^*} w_{\pi'} = 1) \tag{20}$$

$$= \sum_{i \in C^+} \epsilon_i^* \epsilon_i^* = \|\epsilon^*\|_2^2 \tag{21}$$

**Corollary 2.** *For machine user browsing model with top-k consumption and equal attention to the top items, the bound depends on $m$ and $k$. If $m \leq k$, $\langle \epsilon, \epsilon^* \rangle \in [0, m + \frac{(k-m)^2}{n-m}]$. If $m > k$, $\langle \epsilon, \epsilon^* \rangle \in [0, \frac{k^2}{m}]$*

With MU, the upper bound of EE-R can be calculated by equation 6.
If $m \leq k$, $\|\epsilon^*\|_2^2$ becomes

$$\sum_{i=C^+} 1^2 + \sum_{i=C^-} (\frac{k-m}{n-m})^2 = m + \frac{(k-m)^2}{n-m} \tag{22}$$

If $m > k$, $\|\epsilon^*\|_2^2$ becomes

$$\sum_{i=C^+} (\frac{k}{m})^2 + \sum_{i=C^-} 0^2 = \frac{k^2}{m} \tag{23}$$

Therefore, depending on $m$ and $k$, per query $q$, we calculate a normalized EE-R

$$\overline{\text{EE-R}}_q = \begin{cases} \langle \epsilon, \epsilon^* \rangle / (m + \frac{(k-m)^2}{n-m}) & (m \leq k) \\ \langle \epsilon, \epsilon^* \rangle / (\frac{k^2}{m}) & (m > k) \end{cases} \in [0, 1] \tag{24}$$

### C.3 NORMALIZATION OF EU

Theoretically obtaining a true upper bound of a utility of a RAG model is challenging. Therefore, we approximate the true upper bound by the maximum of the empirically obtained utilities given a fixed RAG model $\langle \mathcal{S}, \mathcal{G} \rangle$ with a stochastic ranking sampler.

Recall that the string utility $u_\pi = \mu_u(y, \mathcal{G}(\phi_p(x, \pi)))$ is a utility of a RAG model with a sampled ranking $\pi \in \sigma$. Let $\sigma_\alpha$ denote a sample of rankings with temperature parameter set to $\alpha$. Also, let $\sigma^*$ denote the set of sampled permutations (rankings) from the oracle stochastic retriever, a policy that always places relevant items above non-relevant items; thus the oracle generates $m!(n-m)!$ number of unique optimal permutations.

To obtain an approximated upper bound of the utility $u_{max}$, when the runs of the experiments were run with $\alpha = (1, 2, 4, 8)$, we take the maximum over all samples,

$$u_{max} = max\left(\{u_\pi\}_{\pi \in \sigma_1} \cup \{u_\pi\}_{\pi \in \sigma_2} \cup \{u_\pi\}_{\pi \in \sigma_4} \cup \{u_\pi\}_{\pi \in \sigma_8} \cup \{u_\pi\}_{\pi \in \sigma^*}\right) \tag{25}$$

With $u_{max}$, we *max-normalize* the EU. Since $\frac{1}{N} \sum_{\pi \in \sigma} \frac{u_{\pi}}{u_{max}}$ is the same as $(\frac{1}{N} \sum_{\pi \in \sigma} u_{\pi})/u_{max}$, per query $q$, we get a normalized EU

$$\overline{\text{EU}}_q = \frac{\text{EU}_q}{u_{max}} \quad \in [0, 1] \tag{26}$$

Normalization of EU is done to get the percentage of closeness to the optimal utility as all the utility values are scaled relative to the maximum value. In other words, the normalized EU value indicates how close the EU is to the maximum utility that the RAG system can get to.

This is straightforward for *higher-the-better* metrics, such as ROUGE and Accuracy. However, for *lower-the-better* metrics such as MAE, we convert the scores to *higher-the-better* by subtracting the scores from the true metric upper bound. This allows us to perform the same normalization operation and have the same interpretation of the normalized metric.

# D    Effect of $\alpha$ on Metrics

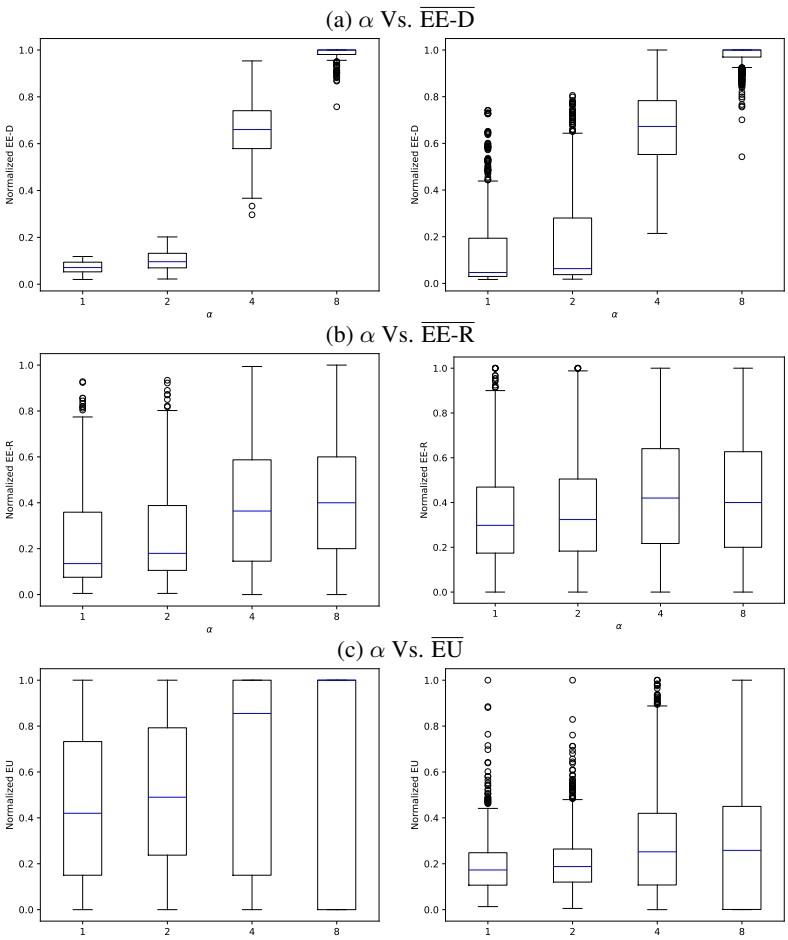

Figure 6: Effect of a temperature parameter $\alpha$ on the (a) disparity of rankings, (b) ranking quality, and (c) generation quality of RAG. The left column is the results on the LaMP task 1 and the right column is on the LaMP task 4, each corresponding to a classification and text generation task, respectively. RAG model is configured with Contriever and Flan-T5-Base for all six figures. Each data point represents the value of the associated metric for one query.

When the three types of plots are observed together, we can infer some interesting observations. In general, we see positive relationships between increasing $\alpha$ and both ranking quality and utility. This implies that we can generally expect a tradeoff between both fairness and ranking quality, as well as between fairness and utility.

However, we can expect some edge cases. For instance, in LaMP-1 (left column of the figure), difference in $\overline{\text{EU}}$ when $\alpha = 4$ and $\alpha = 8$ is not large, and we can see that even the lower quartile of the utility is increased when $\alpha$ is set to 4 (Figure 6c). This can possibly mean that in LaMP-1, the RAG model can maintain considerable utility when the disparity is roughly in the range of $[0.6, 0.8]$. Similar observation can be made for the LaMP-4 (right column of the figure), except that the $\overline{\text{EE-R}}$ is higher when $\alpha = 4$ than when $\alpha = 8$ (Figure 6b). This indicates that the deterministic retriever does not always provide the maximum ranking quality, and the retriever can sometimes provide higher ranking quality by being more fair, ultimately maintaining considerable or higher utility (similar $\overline{\text{EU}}$ when $\alpha$=4 and when $\alpha$=8) at the same time.

With this preliminary observations in mind, we could delve deeper into the relationships between fairness, ranking quality, and utility, by visualizing and quantifying the combined results from all the runs ($\alpha = 1, 2, 4, 8$).

# E INTEGRATING FAIR RANKINGS INTO A RAG SYSTEM IN A PRODUCTION ENVIRONMENT

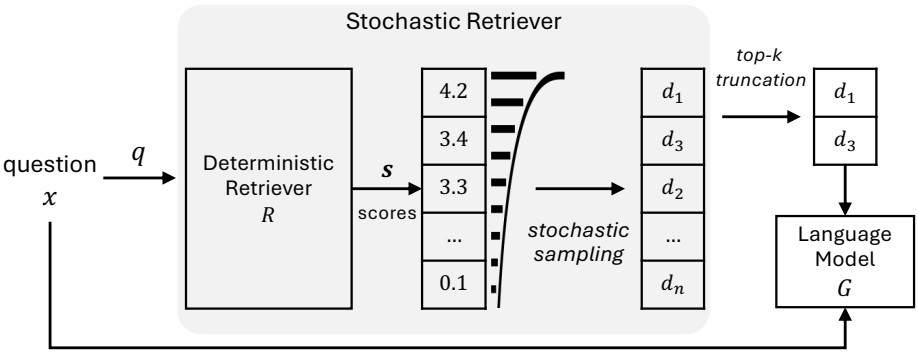

Figure 7: Designing a RAG system that incorporates stochastic fair rankings involves using stochastic sampling, where $N$ can be set to 1 to provide a single ranking to the language model. This can result in a different ranking compared to the one in Figure 1, exposing $d_1$ and $d_3$ to the language model. This paper is focusing on the evaluation of this system as depicted in the Figure 2. The query and prompt generators are omitted in the figure for brevity.

# F FAIRNESS VS. RANKING QUALITY

## F.1 VISUALIZATION OF EE-D VS. EE-R OF FLANT5SMALL

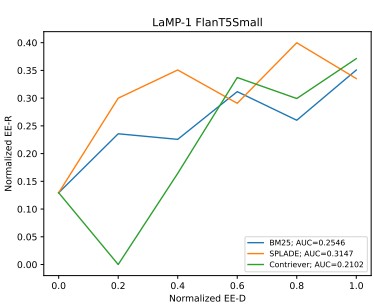

Figure 8: LaMP 1

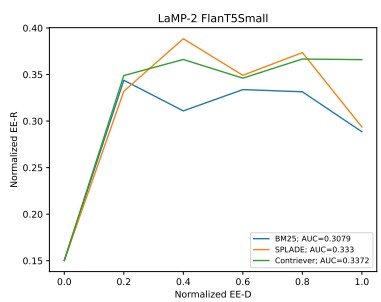

Figure 9: LaMP 2

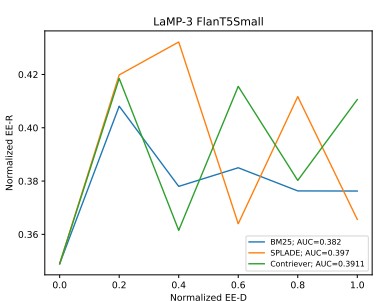

Figure 10: LaMP 3

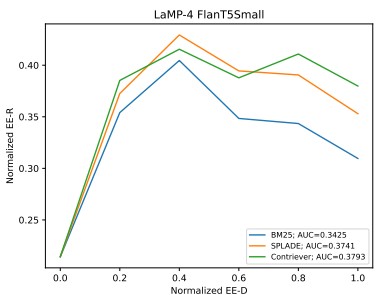

Figure 11: LaMP 4

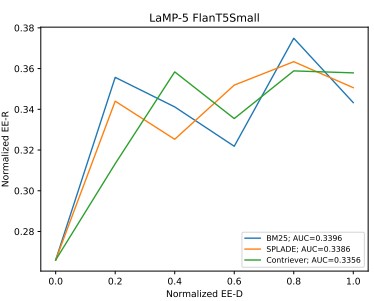

Figure 12: LaMP 5

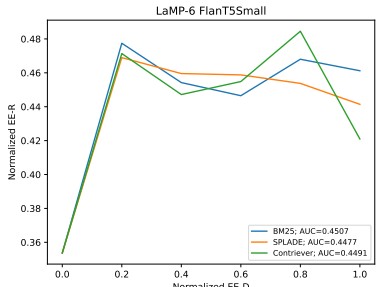

Figure 13: LaMP 6

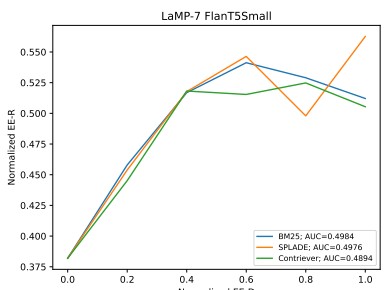

Figure 14: LaMP 7

## F.2 Visualization of EE-D Vs. EE-R of FlanT5Base

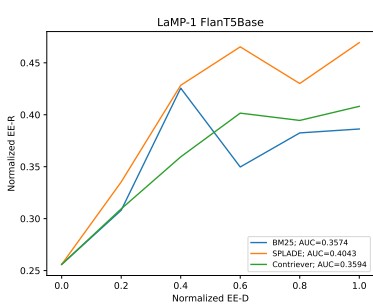

Figure 15: LaMP 1

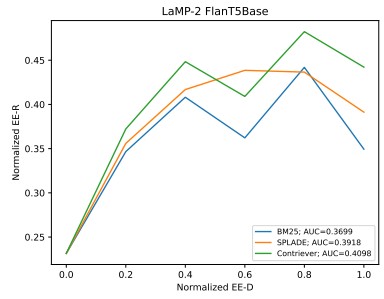

Figure 16: LaMP 2

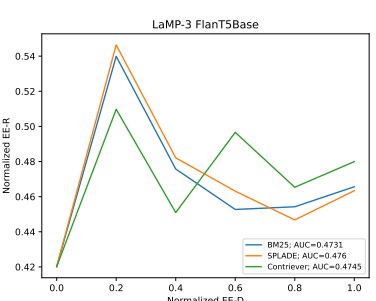

Figure 17: LaMP 3

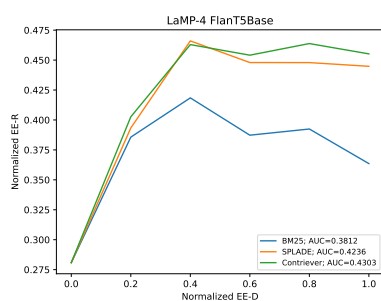

Figure 18: LaMP 4

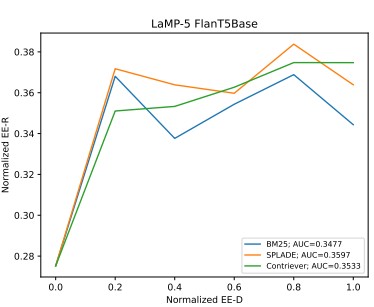

Figure 19: LaMP 5

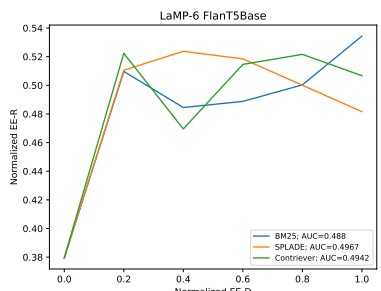

Figure 20: LaMP 6

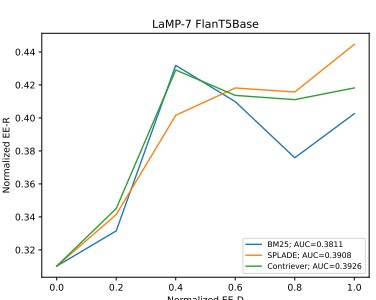

Figure 21: LaMP 7

## F.3 Visualization of EE-D Vs. EE-R of FlanT5XXL

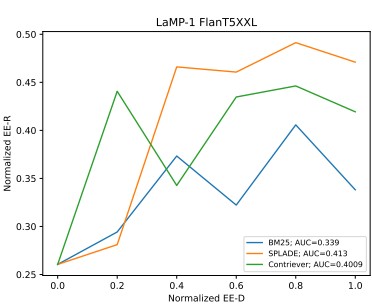

Figure 22: LaMP 1

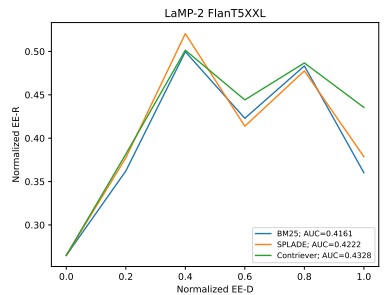

Figure 23: LaMP 2

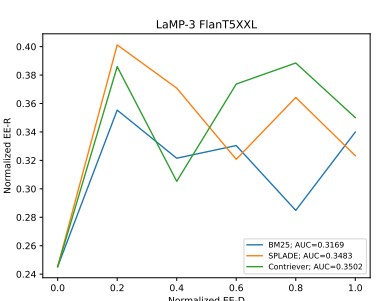

Figure 24: LaMP 3

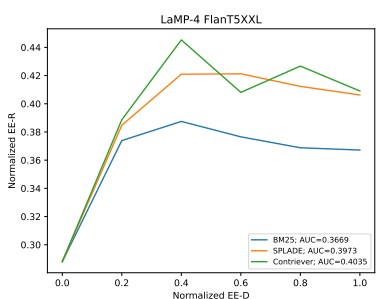

Figure 25: LaMP 4

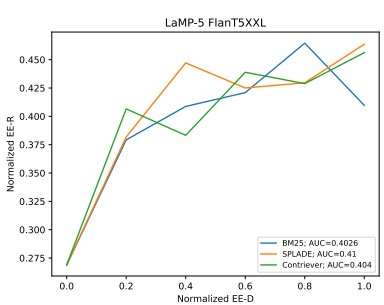

Figure 26: LaMP 5

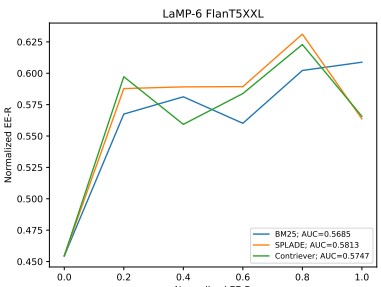

Figure 27: LaMP 6

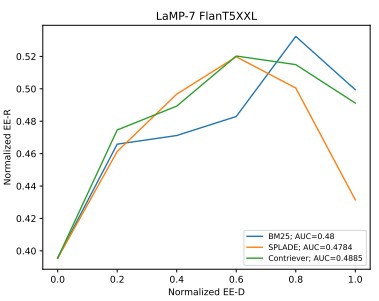

Figure 28: LaMP 7

## F.4 QUANTIFICATION OF FAIRNESS-RANKING QUALITY TRADEOFF

| Task | slope↓ / AUC↑ | slope↓ / AUC↑ | slope↓ / AUC↑ | slope↓ / AUC↑ | slope↓ / AUC↑ | slope↓ / AUC↑ | slope↓ / AUC↑ | slope↓ / AUC↑ | slope↓ / AUC↑ |
|---|---|---|---|---|---|---|---|---|---|
| | FlanT5Small | | | FlanT5Base | | | FlanT5XXL | | |
| | BM25 | SPLADE | Contriever | BM25 | SPLADE | Contriever | BM25 | SPLADE | Contriever |
| LaMP-1 | 0.2113 / 0.2546 | 0.2396 / 0.3147 | 0.2358 / 0.2102 | 0.1546 / 0.3574 | 0.2252 / 0.4043 | 0.1834 / 0.3594 | 0.1358 / 0.3390 | 0.2409 / 0.4130 | 0.1945 / 0.4009 |
| LaMP-2 | 0.1599 / 0.3079 | 0.1863 / 0.3330 | 0.2072 / 0.3372 | 0.1665 / 0.3699 | 0.1899 / 0.3918 | 0.2269 / 0.4098 | 0.1651 / 0.4161 | 0.1834 / 0.4222 | 0.2029 / 0.4328 |
| LaMP-3 | 0.0309 / 0.3820 | 0.0353 / 0.3970 | 0.0501 / 0.3911 | 0.0403 / 0.4731 | 0.0290 / 0.4760 | 0.0555 / 0.4745 | 0.0798 / 0.3169 | 0.1010 / 0.3483 | 0.1338 / 0.3502 |
| LaMP-4 | 0.1271 / 0.3425 | 0.1702 / 0.3741 | 0.1858 / 0.3793 | 0.1054 / 0.3812 | 0.1715 / 0.4236 | 0.1837 / 0.4303 | 0.0882 / 0.3669 | 0.1286 / 0.3973 | 0.1363 / 0.4035 |
| LaMP-5 | 0.1058 / 0.3396 | 0.1014 / 0.3386 | 0.1031 / 0.3356 | 0.0946 / 0.3477 | 0.1144 / 0.3597 | 0.1157 / 0.3533 | 0.2072 / 0.4026 | 0.2071 / 0.4100 | 0.2086 / 0.4040 |
| LaMP-6 | 0.1194 / 0.4507 | 0.1036 / 0.4477 | 0.1090 / 0.4491 | 0.1456 / 0.4880 | 0.1264 / 0.4967 | 0.1438 / 0.4942 | 0.1615 / 0.5685 | 0.1554 / 0.5813 | 0.1467 / 0.5747 |
| LaMP-7 | 0.0962 / 0.4984 | 0.1178 / 0.4976 | 0.0979 / 0.4894 | 0.0744 / 0.3811 | 0.1173 / 0.3908 | 0.0924 / 0.3926 | 0.1174 / 0.4800 | 0.0216 / 0.4784 | 0.0627 / 0.4885 |

Table 3: Values on the left are the gradient of a linear line fit to the data points where x-axis is $\overline{\text{EE-D}}$ and y-axis is $\overline{\text{EE-R}}$. Higher the value, stronger the tradeoff between fairness and ranking quality. Values on the right are the DR-AUC on the disparity-ranking quality ($\overline{\text{EE-D}}$ Vs. $\overline{\text{EE-R}}$) curve. Higher the value, stronger the ranking quality, given consistent tradeoff between fairness and relevance.

# G  RANKING QUALITY VS. UTILITY OF RAG MODELS

## G.1  VISUALIZATION OF THE EE-R VS. EU

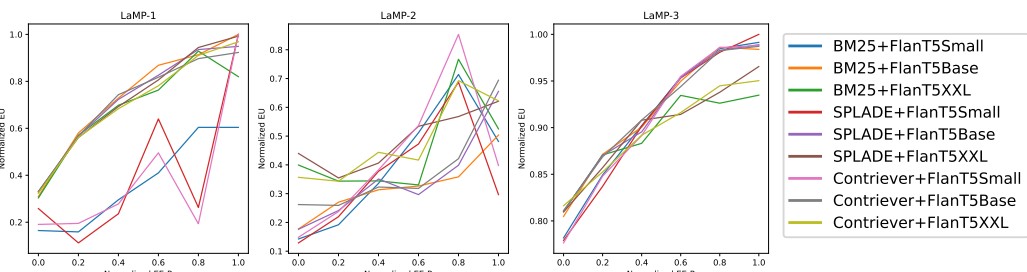

Figure 29: LaMP 1        Figure 30: LaMP 2        Figure 31: LaMP 3

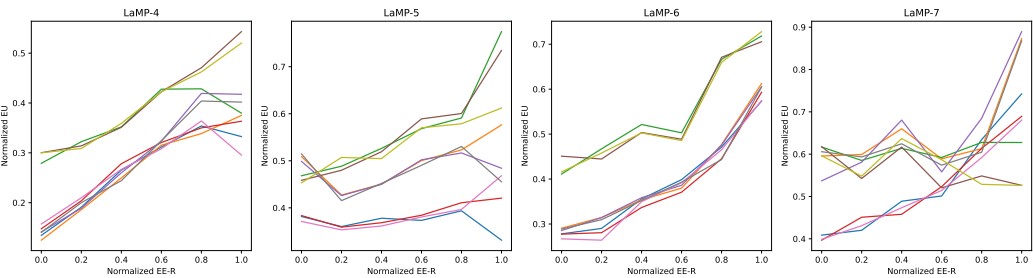

Figure 32: LaMP 4      Figure 33: LaMP 5      Figure 34: LaMP 6      Figure 35: LaMP 7

## G.2  QUANTIFICATION OF THE RELATIONSHIP BETWEEN RANKING QUALITY AND UTILITY

| Task | slope↓ / AUC↑ | slope↓ / AUC↑ | slope↓ / AUC↑ | slope↓ / AUC↑ | slope↓ / AUC↑ | slope↓ / AUC↑ | slope↓ / AUC↑ | slope↓ / AUC↑ | slope↓ / AUC↑ |
|---|---|---|---|---|---|---|---|---|---|
| | BM25 | | | SPLADE | | | Contriever | | |
| | FlanT5Small | FlanT5Base | FlanT5XXL | FlanT5Small | FlanT5Base | FlanT5XXL | FlanT5Small | FlanT5Base | FlanT5XXL |
| LaMP-1 | 0.3813 / 0.3705 | 0.8250 / 0.7491 | 0.7612 / 0.7044 | 0.1434 / 0.3760 | 0.7783 / 0.7382 | 0.7454 / 0.7330 | 0.1864 / 0.3514 | 0.7661 / 0.7304 | 0.7089 / 0.7166 |
| LaMP-2 | 0.5543 / 0.4130 | 0.2858 / 0.3218 | 0.2646 / 0.4495 | 0.5539 / 0.3941 | 0.3310 / 0.3407 | 0.1979 / 0.4787 | 0.6908 / 0.4567 | 0.2325 / 0.3600 | 0.3342 / 0.4772 |
| LaMP-3 | 0.2500 / 0.9153 | 0.2134 / 0.9204 | 0.1633 / 0.8974 | 0.2544 / 0.9124 | 0.2061 / 0.9203 | 0.1737 / 0.9009 | 0.2538 / 0.9125 | 0.2025 / 0.9205 | 0.1568 / 0.8977 |
| LaMP-4 | 0.2708 / 0.2711 | 0.2881 / 0.2679 | 0.1947 / 0.3720 | 0.2638 / 0.2817 | 0.3272 / 0.2942 | 0.2193 / 0.3961 | 0.2264 / 0.2751 | 0.2986 / 0.2887 | 0.2190 / 0.3926 |
| LaMP-5 | -0.0125 / 0.3723 | -0.0307 / 0.4888 | 0.1673 / 0.5591 | -0.0035 / 0.3849 | -0.0160 / 0.4773 | 0.2044 / 0.5570 | 0.0090 / 0.3823 | -0.0474 / 0.4746 | 0.1590 / 0.5387 |
| LaMP-6 | 0.2575 / 0.3883 | 0.2690 / 0.3953 | 0.3078 / 0.5445 | 0.2522 / 0.3737 | 0.2735 / 0.3960 | 0.2780 / 0.5372 | 0.2790 / 0.3783 | 0.2620 / 0.3891 | 0.3053 / 0.5356 |
| LaMP-7 | 0.3229 / 0.5240 | 0.0889 / 0.6391 | 0.0157 / 0.6082 | 0.2781 / 0.5177 | 0.2029 / 0.6435 | -0.0623 / 0.5601 | 0.2713 / 0.5101 | 0.0788 / 0.6266 | -0.0466 / 0.5723 |

Table 4: Values on the left are the gradient of a linear line fit to the data points where x-axis is $\overline{\text{EE-R}}$ and y-axis is $\overline{\text{EU}}$. Higher the value, stronger the tradeoff between retrieval quality and generation quality. Values on the right are the RU-AUC on the ranking quality-utility ($\overline{\text{EE-R}}$ Vs. $\overline{\text{EU}}$) curve. Higher the value, stronger the general end-performance of a RAG model when every level of relevance is considered.

# H ITEM-FAIRNESS VS. UTILITY OF RAG MODELS

## H.1 VISUALIZATION OF EE-D VS. EU

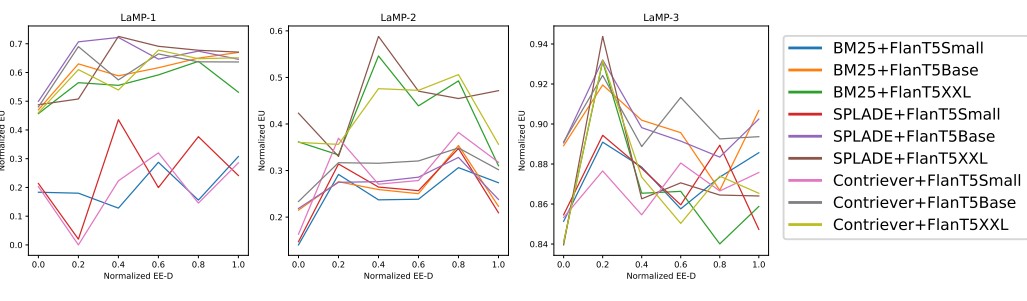

Figure 36: LaMP 1     Figure 37: LaMP 2     Figure 38: LaMP 3

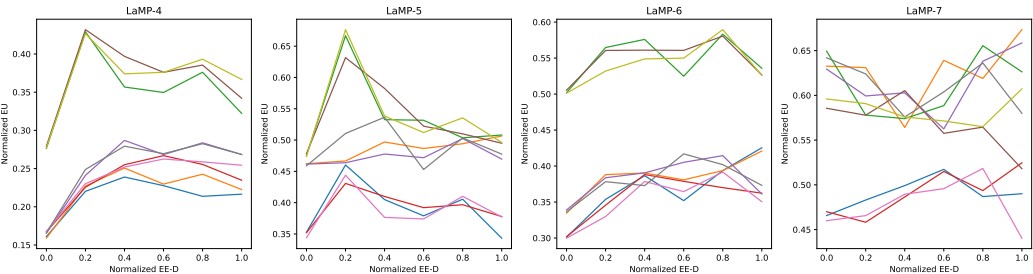

Figure 39: LaMP 4     Figure 40: LaMP 5     Figure 41: LaMP 6     Figure 42: LaMP 7

## H.2 QUANTIFICATION OF FAIRNESS-UTILITY TRADEOFF

| Task | slope↓ / AUC↑ | slope↓ / AUC↑ | slope↓ / AUC↑ | slope↓ / AUC↑ | slope↓ / AUC↑ | slope↓ / AUC↑ | slope↓ / AUC↑ | slope↓ / AUC↑ | slope↓ / AUC↑ |
|------|---|---|---|---|---|---|---|---|---|
| | BM25 | | | SPLADE | | | Contriever | | |
| | FlanT5Small | FlanT5Base | FlanT5XXL | FlanT5Small | FlanT5Base | FlanT5XXL | FlanT5Small | FlanT5Base | FlanT5XXL |
| LaMP-1 | 0.0693 / 0.1994 | 0.2254 / 0.6110 | 0.1673 / 0.5688 | 0.0851 / 0.2519 | 0.1998 / 0.6644 | 0.2385 / 0.6362 | 0.0456 / 0.1866 | 0.2016 / 0.6248 | 0.2413 / 0.6061 |
| LaMP-2 | 0.1295 / 0.2561 | 0.0740 / 0.2717 | 0.0600 / 0.4294 | 0.1119 / 0.2722 | 0.0637 / 0.2786 | 0.0626 / 0.4582 | 0.1683 / 0.3082 | 0.0870 / 0.3139 | 0.1002 / 0.4338 |
| LaMP-3 | 0.0259 / 0.8738 | -0.0012 / 0.8964 | 0.0101 / 0.8706 | 0.0118 / 0.8744 | 0.0014 / 0.9003 | 0.0280 / 0.8786 | 0.0198 / 0.8685 | 0.0063 / 0.9022 | 0.0264 / 0.8765 |
| LaMP-4 | 0.0606 / 0.2178 | 0.0789 / 0.2282 | 0.0734 / 0.3622 | 0.0937 / 0.2408 | 0.1245 / 0.2594 | 0.0895 / 0.3802 | 0.1016 / 0.2429 | 0.1209 / 0.2595 | 0.1051 / 0.3784 |
| LaMP-5 | 0.0293 / 0.3995 | 0.0445 / 0.4857 | 0.0363 / 0.5453 | 0.0437 / 0.3989 | 0.0266 / 0.4763 | 0.0320 / 0.5466 | 0.0533 / 0.3930 | 0.0300 / 0.4943 | 0.0432 / 0.5495 |
| LaMP-6 | 0.1177 / 0.3698 | 0.0753 / 0.3861 | 0.0521 / 0.5535 | 0.0806 / 0.3629 | 0.0623 / 0.3888 | 0.0561 / 0.5558 | 0.0823 / 0.3580 | 0.0597 / 0.3849 | 0.0616 / 0.5468 |
| LaMP-7 | 0.0215 / 0.4928 | 0.0392 / 0.6212 | 0.0854 / 0.6068 | 0.0718 / 0.4901 | 0.0256 / 0.6094 | -0.0668 / 0.5714 | 0.0324 / 0.4839 | -0.0151 / 0.6101 | -0.0114 / 0.5809 |

Table 5: Values on the left are the gradient of a linear line fit to the data points where x-axis is $\overline{\text{EE-D}}$ and y-axis is $\overline{\text{EU}}$. Higher the value, stronger the tradeoff between item-fairness and generation quality. Values on the right are the DU-AUC on the disparity-utility ($\overline{\text{EE-D}}$ Vs. $\overline{\text{EU}}$) curve. Higher the value, stronger the general end-performance of a RAG model when every level of fairness is considered.

# I    PERFORMANCE OF RAG MODELS WITH VARYING FAIRNESS LEVELS

| Model (baseline utility) | Fairness Intervals | | | | |
|---|---|---|---|---|---|
| | [0.0, 0.2) | [0.2, 0.4) | [0.4, 0.6) | [0.6, 0.8) | [0.8, 1.0) |
| BM25+FlanT5Small (0.308) | -0.12 | -0.13 | -0.18 | -0.02 | -0.15 |
| BM25+FlanT5Base (0.670) | -0.20 | -0.04 | -0.08 | -0.05 | -0.02 |
| BM25+FlanT5XXL (0.531) | -0.07 | +0.03 | +0.02 | +0.06 | +0.11 |
| SPLADE+FlanT5Small (0.241) | -0.03 | -0.22 | +0.19 | -0.04 | +0.14 |
| SPLADE+FlanT5Base (0.646) | -0.15 | +0.06 | +0.08 | 0.00 | +0.03 |
| SPLADE+FlanT5XXL (0.671) | -0.18 | -0.16 | +0.05 | +0.02 | +0.01 |
| Contriever+FlanT5Small (0.286) | -0.08 | -0.29 | -0.06 | +0.03 | -0.14 |
| Contriever+FlanT5Base (0.637) | -0.16 | +0.05 | -0.06 | +0.03 | 0.00 |
| Contriever+FlanT5XXL (0.651) | -0.19 | -0.04 | -0.11 | +0.03 | 0.00 |

Table 6: LaMP-1

| Model (baseline utility) | Fairness Intervals | | | | |
|---|---|---|---|---|---|
| | [0.0, 0.2) | [0.2, 0.4) | [0.4, 0.6) | [0.6, 0.8) | [0.8, 1.0) |
| BM25+FlanT5Small (0.274) | -0.13 | +0.02 | -0.04 | -0.04 | +0.03 |
| BM25+FlanT5Base (0.223) | -0.01 | +0.05 | +0.04 | +0.03 | +0.13 |
| BM25+FlanT5XXL (0.310) | +0.05 | +0.02 | +0.24 | +0.13 | +0.18 |
| SPLADE+FlanT5Small (0.209) | -0.06 | +0.10 | +0.06 | +0.05 | +0.14 |
| SPLADE+FlanT5Base (0.238) | -0.02 | +0.04 | +0.04 | +0.05 | +0.09 |
| SPLADE+FlanT5XXL (0.472) | -0.05 | -0.14 | +0.12 | 0.00 | -0.02 |
| Contriever+FlanT5Small (0.318) | -0.15 | +0.05 | -0.05 | -0.04 | +0.06 |
| Contriever+FlanT5Base (0.302) | -0.07 | +0.02 | +0.01 | +0.02 | +0.05 |
| Contriever+FlanT5XXL (0.356) | 0.00 | 0.00 | +0.12 | +0.12 | +0.15 |

Table 7: LaMP-2

| Model (baseline utility) | Fairness Intervals | | | | |
|---|---|---|---|---|---|
| | [0.0, 0.2) | [0.2, 0.4) | [0.4, 0.6) | [0.6, 0.8) | [0.8, 1.0) |
| BM25+FlanT5Small (0.886) | -0.03 | +0.01 | -0.01 | -0.03 | -0.01 |
| BM25+FlanT5Base (0.907) | -0.02 | +0.01 | 0.00 | -0.01 | -0.04 |
| BM25+FlanT5XXL (0.859) | -0.02 | +0.07 | +0.01 | +0.01 | -0.02 |
| SPLADE+FlanT5Small (0.847) | +0.01 | +0.05 | +0.03 | +0.01 | +0.04 |
| SPLADE+FlanT5Base (0.902) | -0.01 | +0.03 | 0.00 | -0.01 | -0.02 |
| SPLADE+FlanT5XXL (0.864) | -0.02 | +0.08 | 0.00 | +0.01 | 0.00 |
| Contriever+FlanT5Small (0.876) | -0.02 | 0.00 | -0.02 | 0.00 | -0.01 |
| Contriever+FlanT5Base (0.894) | 0.00 | +0.03 | 0.00 | +0.02 | 0.00 |
| Contriever+FlanT5XXL (0.865) | -0.02 | +0.07 | +0.01 | -0.02 | +0.01 |

Table 8: LaMP-3

| Model (baseline utility) | Fairness Intervals | | | | |
| --- | --- | --- | --- | --- | --- |
| | [0.0, 0.2) | [0.2, 0.4) | [0.4, 0.6) | [0.6, 0.8) | [0.8, 1.0] |
| BM25+FlanT5Small (0.217) | -0.06 | 0.00 | +0.02 | +0.01 | 0.00 |
| BM25+FlanT5Base (0.223) | -0.06 | 0.00 | +0.03 | +0.01 | +0.02 |
| BM25+FlanT5XXL (0.322) | -0.05 | +0.11 | +0.03 | +0.03 | +0.05 |
| SPLADE+FlanT5Small (0.235) | -0.07 | -0.01 | +0.02 | +0.03 | +0.02 |
| SPLADE+FlanT5Base (0.268) | -0.10 | -0.03 | +0.02 | 0.00 | +0.02 |
| SPLADE+FlanT5XXL (0.342) | -0.06 | +0.09 | +0.05 | +0.03 | +0.04 |
| Contriever+FlanT5Small (0.254) | -0.09 | -0.02 | 0.00 | +0.01 | 0.00 |
| Contriever+FlanT5Base (0.268) | -0.10 | -0.02 | +0.01 | 0.00 | +0.01 |
| Contriever+FlanT5XXL (0.367) | -0.09 | +0.06 | +0.01 | +0.01 | +0.03 |

Table 9: LaMP-4

| Model (baseline utility) | Fairness Intervals | | | | |
| --- | --- | --- | --- | --- | --- |
| | [0.0, 0.2) | [0.2, 0.4) | [0.4, 0.6) | [0.6, 0.8) | [0.8, 1.0] |
| BM25+FlanT5Small (0.343) | +0.01 | +0.12 | +0.06 | +0.04 | +0.06 |
| BM25+FlanT5Base (0.507) | -0.04 | -0.04 | -0.01 | -0.02 | -0.01 |
| BM25+FlanT5XXL (0.508) | -0.03 | +0.16 | +0.02 | +0.02 | 0.00 |
| SPLADE+FlanT5Small (0.378) | -0.03 | +0.05 | +0.03 | +0.01 | +0.02 |
| SPLADE+FlanT5Base (0.470) | -0.01 | -0.01 | +0.01 | 0.00 | +0.03 |
| SPLADE+FlanT5XXL (0.495) | -0.02 | +0.14 | +0.09 | +0.03 | +0.01 |
| Contriever+FlanT5Small (0.377) | -0.03 | +0.07 | 0.00 | 0.00 | +0.03 |
| Contriever+FlanT5Base (0.478) | -0.02 | +0.03 | +0.06 | -0.02 | +0.03 |
| Contriever+FlanT5XXL (0.496) | -0.02 | +0.18 | +0.04 | +0.02 | +0.04 |

Table 10: LaMP-5

| Model (baseline utility) | Fairness Intervals | | | | |
| --- | --- | --- | --- | --- | --- |
| | [0.0, 0.2) | [0.2, 0.4) | [0.4, 0.6) | [0.6, 0.8) | [0.8, 1.0] |
| BM25+FlanT5Small (0.425) | -0.12 | -0.07 | -0.04 | -0.07 | -0.03 |
| BM25+FlanT5Base (0.421) | -0.09 | -0.03 | -0.03 | -0.04 | -0.03 |
| BM25+FlanT5XXL (0.536) | -0.03 | +0.03 | +0.04 | -0.01 | +0.05 |
| SPLADE+FlanT5Small (0.362) | -0.06 | -0.02 | +0.03 | +0.02 | +0.01 |
| SPLADE+FlanT5Base (0.361) | -0.02 | +0.02 | +0.03 | +0.04 | +0.05 |
| SPLADE+FlanT5XXL (0.527) | -0.02 | +0.03 | +0.03 | +0.03 | +0.05 |
| Contriever+FlanT5Small (0.351) | -0.05 | -0.02 | +0.03 | +0.01 | +0.04 |
| Contriever+FlanT5Base (0.373) | -0.04 | +0.01 | 0.00 | +0.04 | +0.03 |
| Contriever+FlanT5XXL (0.526) | -0.02 | +0.01 | +0.02 | +0.02 | +0.06 |

Table 11: LaMP-6

| Model (baseline utility) | Fairness Intervals | | | | |
| --- | --- | --- | --- | --- | --- |
| | [0.0, 0.2) | [0.2, 0.4) | [0.4, 0.6) | [0.6, 0.8) | [0.8, 1.0] |
| BM25+FlanT5Small (0.490) | -0.02 | -0.01 | +0.01 | +0.03 | 0.00 |
| BM25+FlanT5Base (0.673) | -0.04 | -0.04 | -0.11 | -0.03 | -0.05 |
| BM25+FlanT5XXL (0.626) | +0.02 | -0.05 | -0.05 | -0.04 | +0.03 |
| SPLADE+FlanT5Small (0.525) | -0.05 | -0.07 | -0.04 | -0.01 | -0.03 |
| SPLADE+FlanT5Base (0.659) | -0.03 | -0.06 | -0.06 | -0.10 | -0.02 |
| SPLADE+FlanT5XXL (0.518) | +0.07 | +0.06 | +0.09 | +0.04 | +0.05 |
| Contriever+FlanT5Small (0.440) | +0.02 | +0.03 | +0.05 | +0.06 | +0.08 |
| Contriever+FlanT5Base (0.580) | +0.06 | +0.04 | 0.00 | +0.02 | +0.06 |
| Contriever+FlanT5XXL (0.607) | -0.01 | -0.02 | -0.03 | -0.04 | -0.04 |

Table 12: LaMP-7

# J DATA STATISTICS

## J.1 LAMP DATA STATISTICS FOR FLAN-T5-SMALL

| Dataset | #queries | Avg # Docs (Std) | Avg # Pos Labels (Std) | Avg % Pos Labels |
|---------|----------|------------------|------------------------|------------------|
| LaMP-1 | 51 | 123.51 (82.66) | 9.08 (11.63) | 9.53 |
| LaMP-2 | 192 | 52.81 (46.21) | 7.98 (9.64) | 22.53 |
| LaMP-3 | 311 | 189.82 (134.33) | 65.88 (95.77) | 34.28 |
| LaMP-4 | 833 | 192.19 (195.28) | 40.72 (61.82) | 27.1 |
| LaMP-5 | 826 | 106.06 (71.47) | 26.18 (31.1) | 24.83 |
| LaMP-6 | 760 | 86.0 (52.66) | 27.78 (29.22) | 35.92 |
| LaMP-7 | 365 | 19.36 (18.4) | 8.23 (10.38) | 45.48 |

Table 13: LaMP data statistics for Flan-T5-Small after filtering for fairness evaluation.

## J.2 LAMP DATA STATISTICS FOR FLAN-T5-BASE

| Dataset | #queries | Avg # Docs (Std) | Avg # Pos Labels (Std) | Avg % Pos Labels |
|---------|----------|------------------|------------------------|------------------|
| LaMP-1 | 232 | 102.86 (61.88) | 20.07 (22.78) | 22.49 |
| LaMP-2 | 280 | 45.58 (42.0) | 10.45 (12.56) | 29.87 |
| LaMP-3 | 378 | 185.32 (128.43) | 73.82 (85.44) | 41.19 |
| LaMP-4 | 827 | 186.98 (193.52) | 49.79 (68.91) | 31.57 |
| LaMP-5 | 759 | 105.62 (69.56) | 26.09 (31.21) | 25.71 |
| LaMP-6 | 783 | 86.18 (52.97) | 30.11 (31.28) | 38.65 |
| LaMP-7 | 211 | 21.72 (16.09) | 6.96 (10.62) | 33.02 |

Table 14: LaMP data statistics for Flan-T5-Base after filtering for fairness evaluation.

## J.3 LAMP DATA STATISTICS FOR FLAN-T5-XXL

| Dataset | #queries | Avg # Docs (Std) | Avg # Pos Labels (Std) | Avg % Pos Labels |
|---------|----------|------------------|------------------------|------------------|
| LaMP-1 | 264 | 111.66 (69.45) | 25.12 (33.96) | 23.35 |
| LaMP-2 | 105 | 44.66 (42.82) | 11.32 (15.61) | 36.74 |
| LaMP-3 | 182 | 198.06 (151.09) | 41.86 (59.52) | 22.19 |
| LaMP-4 | 842 | 198.0 (200.82) | 54.07 (73.34) | 30.96 |
| LaMP-5 | 511 | 104.18 (68.73) | 23.1 (38.3) | 23.39 |
| LaMP-6 | 730 | 85.93 (52.46) | 34.89 (35.54) | 43.88 |
| LaMP-7 | 151 | 20.6 (16.39) | 8.58 (12.01) | 42.7 |

Table 15: LaMP data statistics for Flan-T5-XXL after filtering for fairness evaluation.

## K   IMPLEMENTATION DETAILS

**BM25**:

- Adapted from: `https://github.com/dorianbrown/rank_bm25/tree/master`

**SPLADE**:

- `https://huggingface.co/naver/splade_v2_max`

**Contriever**:

- `https://huggingface.co/facebook/contriever`

**Flan-T5 Family**:

- `https://huggingface.co/google/flan-t5-small`
- `https://huggingface.co/google/flan-t5-base`
- `https://huggingface.co/google/flan-t5-xxl`

The RAG model inferences were performed on NVIDIA A6000 GPUs with 48GB of VRAM.