# OpenReview forum: "Towards Fair RAG: On the Impact of Fair Ranking in Retrieval-Augmented Generation"
_ICLR.cc/2025/Conference — ICLR 2025 Conference Withdrawn Submission_

### Official Review · Reviewer_wyXA · 2024-10-29

**Soundness:** 2
**Presentation:** 3
**Contribution:** 2
**Rating:** 5
**Confidence:** 4

**Summary:**

This paper presents an interesting phenomenon in RAG: providing more equitable exposure for different items in RAG leads to improved performance outcomes. The authors also show that there is a general trend of a tradeoff between ensuring fairness and maintaining system effectiveness.

I have some concerns about whether this qualifies as a definition of fairness rather than as a form of bias. The bias could originate from the close form of documents and the retriever. Though the paper presents an interesting problem, it would be more convincing from a deeper exploration of the underlying causes of this bias and from proposing potential methods to address the issue.

**Strengths:**

S1: This paper presents an interesting scenario: providing more equitable exposure for different items in RAG leads to improved performance outcomes.

S2: The authors conduct extensive experiments to show a general trend of a tradeoff between ensuring fairness and maintaining system effectiveness.

**Weaknesses:**

W1: Does this qualify as a definition of fairness? The outcome is equaliable does not necessarily be fair. When it comes to fairness [1,2], it leans more toward a subjective goal: for instance, even if retrieval achieves 100% accuracy, it may still conflict with human values, such as when certain categories receive less exposure. However, as for bias, it mainly cares about the final utility (objective). As for this setting, It seems it is a form of bias because the final RAG goal is to gain more utility. The issue you've highlighted may stem from certain biases. When equal exposure is given to different documents, overall utility tends to improve. This might be because some documents, though scoring similarly to the query, contribute disproportionately to the final utility, while others do not. Equally retrieval of them may somehow improve the expected utility (because some items may then be exposed to LLMs). This discrepancy could be due to a mismatch between the retriever and LLMs or other underlying factors. I encourage the author to find a deep reason behind this problem.

[1] Sunhao Dai, Chen Xu, Shicheng Xu, Liang Pang, Zhenhua Dong, and Jun Xu. 2024. Bias and Unfairness in Information Retrieval Systems: New Challenges in the LLM Era. In Proceedings of the 30th ACM SIGKDD Conference on Knowledge Discovery and Data Mining (KDD '24). Association for Computing Machinery, New York, NY, USA, 6437–6447.

[2] Ferrara, E. (2023). Fairness and bias in artificial intelligence: A brief survey of sources, impacts, and mitigation strategies. Sci, 6(1), 3.


W2: The generation models are not large enough. It is better to conduct experiments on 7B-sized models such as Llama. RAG is more widely used in such LLMs.


W3: Including some case studies would help readers understand which types of documents require fairer exposure. Alternatively, conducting experiments to identify which documents need more visibility could provide insights into the underlying causes of this bias.

**Questions:**

See the above comments

---

### Official Review · Reviewer_BuKW · 2024-10-29

**Soundness:** 3
**Presentation:** 2
**Contribution:** 2
**Rating:** 3
**Confidence:** 3

**Summary:**

The paper focuses on investigating the integration of fair ranking methods into Retrieval-Augmented Generation (RAG) systems. They conduct evaluations for item-side fairness, which aims to ensure equitable exposure for relevant item providers in the retrieved rankings used by RAG systems. Their findings reveal that incorporating fair rankings can maintain or even improve the generation quality of RAG systems compared to traditional methods.

**Strengths:**

1. The research problem is interesting and important, which is essential for the responsible deployment of RAG systems.
2. Experiments results show that fair rankings can maintain or even improve the generation quality of RAG.

**Weaknesses:**

1. Section 3 introduces extensive notation and terminology that may be unnecessary, making the content difficult to follow the experimental settings. Simplifying this section by clearly explaining the evaluation settings and metrics without excessive symbols would enhance readability and comprehension.

2. The paper evaluates only one fair ranking method, which limits the generalizability of the findings. Incorporating other item-side fairness ranking methods (e.g., refer to [1]) would strengthen the evaluation and provide a more comprehensive understanding.

3. Despite the claim of publicly releasing the code and dataset in the abstract, they are not available.

[1] Fairness in Recommendation: Foundations, Methods and Applications

**Questions:**

1. Why only choose stochastic retrievers as the fair ranking method?
2. Given the presence of selection bias or position bias of LLMs, documents at different positions in the retrieved ranking may have an unequal influence on the final generated answer. This could mean that Formula (4) does not hold as assumed.

---

### Official Review · Reviewer_6Cxj · 2024-11-04

**Soundness:** 2
**Presentation:** 3
**Contribution:** 2
**Rating:** 3
**Confidence:** 4

**Summary:**

This work investigates the impacts of ranking fairness over the performance of RAG systems. The ranking fairness that is related to the exposure of relevant document is not well discussed in the era of LLM-based RAG. Therefore, this paper leverage the Expected Exposure for item ranking as mesurements to explore the relationships between item-fairness and ranking quality, as well as those between item-fairness and generation quality. The experimental results are discussed to some extend.

**Strengths:**

1.The motivation is clear and the experimental setups is detailed introduced.
2.The paper is well-written.

**Weaknesses:**

1. This paper is a trivial work with incremental contributions, which explores the fairness impact of LLM-based RAG systems. In fact, there are few valuable findings compared to previous studies. Many previous studies found that retrieval diversities (akin to fairness) and position biases (e.g., loss-in-the-middle phenomenon) influence the RAG performance a lot.
2. The experiments are not sufficiently thorough. There are only two main discussions about the relationships among ranking fairness, ranking quality, and generation quality. More detailed analysis and discussions are required for a comprehensive investigation.
3. The experiments conducted on the Flan-T5 family are not convincing to me. Since recent LLM-based RAGs are mostly built over larger and more powerful LLMs, e.g., GPT 4 and LLaMA3.x. They usually demonstrate different performances compared to other pre-trained LMs (e.g., Flan-T5-Small). The impact discussion should mainly based on the LLM-based RAGs.

**Questions:**

None

---

### Official Review · Reviewer_mbfd · 2024-11-08

**Soundness:** 3
**Presentation:** 4
**Contribution:** 3
**Rating:** 8
**Confidence:** 4

**Summary:**

In this paper, the authors investigate the impact of fair ranking on RAG systems.  They conduct systematic evaluations of RAG systems integrated with fair rankings. Based on the experiments, they summarize several key findings, for example, using fair rankings can maintain a high level of generation quality and sometimes it can improve generation quality.

Pros:
-  The problems discussed in the paper is interesting and important.
- The experimental studies and the findings are useful to the research community, although the experiments still have some limitations. For example, only fair exposure is considered.
- The paper is well-written and easy to follow.

Cons:
- Given the truth that long-context modeling has been widely applied in many LLMs, it would be great if the discussion in the paper can be extended to such models. I believe if more results can be fed into LLMs, the fairness problem should be different with the problem studied in the paper.
- More advanced problems should also be considered. For example, the current RAG system has refiner components. More discussion about fairness in these components should be discussed.

**Strengths:**

Pros:
-  The problems discussed in the paper is interesting and important.
- The experimental studies and the findings are useful to the research community, although the experiments still have some limitations. For example, only fair exposure is considered.
- The paper is well-written and easy to follow.

**Weaknesses:**

Cons:
- Given the truth that long-context modeling has been widely applied in many LLMs, it would be great if the discussion in the paper can be extended to such models. I believe if more results can be fed into LLMs, the fairness problem should be different with the problem studied in the paper.
- More advanced problems should also be considered. For example, the current RAG system has refiner components. More discussion about fairness in these components should be discussed.

**Questions:**

NA

---

### Note · Authors · 2024-11-20

I have read and agree with the venue's withdrawal policy on behalf of myself and my co-authors.